# A Review of Potato Salt Tolerance

**DOI:** 10.3390/ijms241310726

**Published:** 2023-06-27

**Authors:** Xue Han, Ruijie Yang, Lili Zhang, Qiaorong Wei, Yu Zhang, Yazhi Wang, Ying Shi

**Affiliations:** College of Agriculture, Northeast Agricultural University, Harbin 150030, China

**Keywords:** potato, salt tolerance, genetic engineering, multiomics

## Abstract

Potato is the world’s fourth largest food crop. Due to limited arable land and an ever-increasing demand for food from a growing population, it is critical to increase crop yields on existing acreage. Soil salinization is an increasing problem that dramatically impacts crop yields and restricts the growing area of potato. One possible solution to this problem is the development of salt-tolerant transgenic potato cultivars. In this work, we review the current potato planting distribution and the ways in which it overlaps with salinized land, in addition to covering the development and utilization of potato salt-tolerant cultivars. We also provide an overview of the current progress toward identifying potato salt tolerance genes and how they may be deployed to overcome the current challenges facing potato growers.

## 1. Introduction

Potato is the fourth most important food crop in the world [1]. The ploidy of potatoes varies significantly in nature, ranging from diploid to hexaploid, but only diploid and tetraploid potatoes are used in modern cultivation [2,3]. Potato tubers contain proteins, starch, fats, reducing sugars, dietary fiber and other major nutrients, and they are rich in mineral elements such as potassium, iron, zinc, calcium, magnesium and vitamins. The content of vitamin C in potato is higher than that of wheat and other staple crops, and the content of protein and reducing sugar is higher than that of many common vegetables [4,5]. In addition to the direct consumption of fresh potatoes, potatoes are also widely used in the animal feed and food processing industries, and fresh potatoes and their processed products contribute to agricultural product trade and the global economy [6,7,8,9,10,11,12]. Compared with other food crops, potato has high adaptability, cold tolerance and drought resistance, resulting in stable yields [13]. These features make potato the only staple crop that can be planted in many areas of the world. Potato also plays a major role in food security for much of the developing world [14]. European and American countries have long consumed potato as a staple food, and Asian countries have increased their consumption more recently. Increased planting of potato in African countries has resulted in improved food security, leading to the United Nations naming 2008 as the ‘International Year of Potato’ [15].

During growth, potato utilizes nutrients stored in its tuber for vegetative propagation, including the bud growth stage, seedling stage, tuber formation stage, tuber growth stage, starch accumulation stage and maturity stage. Potatoes require very specific soil characteristics for ideal growth and yield, including soil with loose structure, good ventilation and high organic matter content [16]. In addition, potatoes are moderately sensitive to salt and grow best when the soil pH ranges from 5.0 to 5.5 [17]. Therefore, in salinized soil, the above-ground parts of salt-sensitive potato cultivars will have delayed seedling emergence and leaf wilting [18], and the underground parts will not take root and produce potatoes [19], resulting in dramatic yield reductions. The exploration of saline tolerance genes may therefore result in large productivity gains for regions with salinized soil.

## 2. Global Potato Planting Area and Consumption

Potatoes were first discovered in the Andes Mountains of South America, but their cultivation rapidly spread across the world due to their strong environmental adaptability and high nutritional value [20]. According to the Food and Agriculture Organization of the United Nations (FAO), potatoes were grown in more than 150 countries and territories around the world in 2014. From 2013 to 2020, the global potato planting area declined slightly, reaching approximately 16,887.04 thousand hectares in 2020. The total potato output fluctuated widely over this time frame, with the lowest output in 2016 (Figure 1A). Asia and Europe represent the two largest potato planting areas in the world. In 2002, Asia surpassed Europe and became the largest potato producer in the world. Within Europe, the majority of potato production occurs in Russia and Ukraine. In 2020, Ukraine’s potato planting area accounted for 7.85% of the world’s total, with a fresh potato output of 20,837,990 tons, ranking third. Russia’s potato growing area accounts for 6.98% of the world’s total, with a total output of 19.6074 million tons, ranking fourth (Figure 2 and Figure 3). Potato-growing countries in Asia include China, India and several others [21]. In 2020, China’s potato growing area accounted for 27.69% of the world’s total, and its fresh food output was 92,800,000 tons, ranking first. India’s potato growing area accounted for 12.15% of the world’s total, with a fresh food output of 48,562,000 tons, ranking second (Figure 2 and Figure 3). As the backbone of potato cultivation in Asia, China has a long history of more than 400 years of potato cultivation. Over time, this cultivation area has gradually expanded. There are now four major potato producing areas in China, including the single-season cropping area of the north, the two-season cropping area in the central plains, the two-season mixed cropping area in the southwest and the winter cropping area in the south. From 2013 to 2020, the growing area of potato in China decreased slightly, and the yield increased moderately (Figure 1B). In 2020, the potato planting area in China reached 4656 thousand hectares, an increase of 3355.2 thousand hectares from 1961 [22]. In the past two years, frequent harsh weather conditions in the northern cropping area combined with lower profits and policy changes have caused a shift in the potato planting area towards the southwest. Despite these changes, the overall potato planting area in China is expected to remain stable, and yield is expected to continue to increase.

## 3. Current Challenges with Salinized Land

According to the FAO, the world had 1.56 billion hectares of farmland in 2020. According to the global saline soil map released by the FAO in October 2021, more than 833 million hectares of this area were affected by salinization, representing more than half of the total farmland area. This worldwide environmental problem has become an important factor hindering the development of agriculture [23,24,25]. Salt stress not only threatens crop productivity but also causes global economic losses of approximately USD 27.3 billion per year [26,27]. Salinized land is widely distributed across the world, with large areas affected in Australia, Canada, the United States, India and other countries [28,29]. China ranks third for global salinized soil [30]. Salinized soils mainly contain a large amount of Cl^−^, SO^42−^, Mg^2+^, Na^+^, K^+^ and Ca^2+^. [31,32,33]. The causes of soil salinization are complex and include a combination of natural factors and human impacts [34,35,36,37]. The Northeast China Plain represents a good example of the salinization process. During spring, this area has high winds and low rainfall, and during winter, it is subject to frequent freeze–thaw weather patterns. These factors result in a steady accumulation of salt in the surface soil, which is exacerbated by human activities, including irrigation with river water and over-grazing [38,39,40,41]. The gradual salinization of cultivated land inevitably decreases cultivation area, leading to economic and food security problems. It is therefore important to create and utilize crop cultivars that are tolerant to salt stress [42,43,44,45].

## 4. Effects of Salt on Growth, Physiology and Biochemistry of Potato

In agricultural production, salt stress is one of the main abiotic stresses affecting potato. Under stress, potato plants experience significant changes in internal physiology and external morphology, leading to reduced tuber quality and yield.

### 4.1. Effect of Salt on Potato Growth and Development

Potato is a vegetative crop whose main economic value is dependent on the quality and yield of potato tubers. Although potato is a somewhat salt-tolerant plant, salt stress directly affects the quality and yield of potato tubers, with salt stress having a more direct impact on potato compared to other plants [46]. Salt stress can cause significant damage during all growth stages of potato. However, potato has the highest demand for nutrients during the tuber expansion stage, and salt stress during this developmental window is particularly damaging. In high-salt soil, salt-sensitive cultivars of potato have reduced seedling emergence, growth and development, and salt stress can aggravate scab disease, which directly affects the commercial value of potato. Studies have shown that there is a significant positive correlation between soil conductivity and salt content, and the higher the salinity, the higher the soil conductivity [47]. When the soil conductivity exceeds 1.95 dSm^−1^, seed potatoes will rot [48]. Salt can also delay the germination of potato seeds [48]. Additionally, salt severely affects root development, and a strong root system is a prerequisite for the robust growth of all plants. Salt can lead to a decreased rooting rate, shorter root length and lower root vitality [49,50]. The poor development of roots and the weakening of water absorption ability hinder the growth of plants at the seedling stage. Furthermore, salt can reduce the number of leaves and cause existing leaves to turn yellow and curled, eventually leading to the wilting and dropping of leaves [51]. The tuber formation stage determines the number of tubers per plant, and salt has been shown to inhibit tuber formation. At high salt levels, the growth of microtubers has been shown to be inhibited, and the number of microtubers decreases with increasing salinity [52]. Reduced photosynthesis of above-ground leaves and nutrient uptake of underground roots caused by salt limit tuber expansion and can reduce dry matter accumulation in potatoes. The presence of salt can eventually lead to a decrease in the number of tubers per plant, average tuber weight and total yield of potato.

### 4.2. Effects of Salt on Potato Physiology and Biochemistry

Salt stress has been shown to not only affect the morphology of potato, but also to interfere with many internal physiological and biochemical processes, causing damage to organs and tissues and greatly reducing the quality of tubers. Due to the competitive absorption of ions, high concentrations of sodium and chloride ions can have antagonistic effects on other ions and perturb the ion balance of plants [53,54]. At the same time, salt increases the permeability of cell membranes and damages the cell membrane structure [55,56].

Excessive accumulation of sodium ions also reduces potassium absorption, which inhibits the function of many enzymes that depend on potassium ions [57]. Reactive oxygen species (ROS) are by-products of aerobic metabolism in plants, including peroxides, superoxides, hydroxyl radicals and singlet oxygen [58]. Normally, the production and removal of reactive oxygen species are balanced, resulting in a relatively low level of ROS. When the salt concentration is too high, the synthesis of antioxidant enzymes increases, which can remove harmful substances, such as reactive oxygen species and free radicals. Antioxidant enzymes, such as superoxide dismutase (SOD), peroxidase (POD), catalase (CAT) and others, are often used as indicators to measure whether potatoes are salt-tolerant. Under high salt, potassium ions are reduced, and the activity of antioxidant enzymes is inhibited. Additionally, CO_2_ fixation during photosynthesis is impaired, which leads to the activation of O_2_ molecules and the production of a large number of reactive oxygen species [23,57,59]. Excessive accumulation of reactive oxygen species can cause oxidative damage to proteins, lipids and DNA, and excessive accumulation of chloride ions in plants leads to reduced chlorophyll content [60,61,62,63]. Additionally, salt reduces soil water potential, reducing water absorption, resulting in osmotic stress, decreased leaf turgor pressure and the closure of stomatal cells [23]. These changes reduce photosynthesis, further impairing normal growth.

Chloride ions also lead to impaired carbohydrate distribution, reduced starch content and reduced quality of tubers [64]. In addition, salt has a significant effect on protein metabolism, inhibiting protein synthesis and promoting its hydrolysis. For example, free proline has been shown to accumulate under salt stress [65,66]. This reduction in protein also affects starch content, causing dramatic reductions in total potato yield.

## 5. Progress toward the Identification of Salt Tolerance Genes in Potato

### 5.1. Advances in Potato Genome Research

Cultivated potato is an autotetraploid, with high genomic heterozygosity, 10 gametes and 35 zygotes. The inheritance pattern of potato is very complex, and the phenotypes of offspring generated by sexual reproduction are unpredictable, severely limiting the rate of genetic improvement [67,68,69]. In agricultural production, potato blocks are used for vegetative propagation to obtain stable genetic characteristics. However, the use of tubers for propagation results in a low reproduction coefficient, large storage and transportation costs and high susceptibility to viral infection and insect infestation [70,71]. Potato breeding using traditional methods is a slow process, and a better understanding of the potato genome could greatly accelerate molecular breeding efforts [72]. In order to overcome the obstacles posed by tetraploid inheritance, potato was domesticated from a tetraploid crop that is propagated by tubers into a diploid crop propagated by seed, making it amenable to genome sequencing and molecular characterization [73].

In 2006, 29 scientific research institutions from 14 countries jointly formed the International Potato Genome Sequencing Collaborative Group (PGSC) to carry out potato genome sequencing and functional annotation [74]. PGSC initially investigated the genome of RH89-039-16 (RH), a diploid material that is closely related to commercial tetraploid potato cultivars. However, due to the limitations of high heterozygosity, low quality of physical maps and high sequencing costs of the diploid potato genome, progress eventually stalled. Dr. Huang Sanwen, a member of the PGSC group, proposed the use of the haploid potato DM1-3 516 R44 (DM), as well as the whole-genome shotgun method and the application of next-generation sequencing technology with low costs to overcome these issues. This new strategy solved the problems of the high complexity and high cost of genome analysis and greatly accelerated progress towards a complete potato haploid genome containing 12 chromosomes and 840 million base pairs. In 2009, the potato genome sequence framework, covering more than 95% of all genes, was completed [75]. In 2011, the functionally annotated potato doubled haploid reference genome was released, providing a reference genome for researchers studying potato at the genomic level. In this study, haploid potato was obtained in vitro by another culture of the diploid material *Solanum tuberosum* group Phureja clone (PI 225669). The homozygous doubled haploid DM was then obtained by chromosome doubling, followed by DNA extraction and genome sequencing. The PGSC also analyzed the genome sequence of the diploid material RH, which had a genome assembly that was only 10% complete [76]. The DM reference genome represents a wealth of data that is critical for the future genetic improvement of potato. However, due to the high heterozygosity of the potato genome and the difference between haplotypes, the information provided by the DM reference genome of dihaploid potato has some important limitations, and additional genomes are needed. The successful construction of high-quality pangenomes of several species, such as soybeans and rice, shows that many different lines are needed to capture the diversity of crop species [77,78].

Wild diploid potato has high levels of genetic diversity and polymorphism, especially in resistance genes. These genes confer resistance to a variety of diseases, including potato late blight. Potato late blight, caused by the *Phytophthora infestans*, causes significant damage to world potato production. Researchers have found dominant resistance genes against late blight (*Rpi* genes) in wild potato species, which have greatly reduced its damage to potato production [79]. The wild diploid potato *Solanum commersonii*, for example, contains genes related to pathogen defense as well as low-temperature tolerance that are missing in cultivated potato. Due to reproductive isolation between the wild diploid potato and cultivated potato, it is difficult to make hybrid crosses that would enable the integration of desirable wild traits, such as stress tolerance and disease resistance [80,81]. In 2015, an Italian research team assembled and published the *Solanum commersonii* genome using second-generation sequencing data, enabling more easy utilization of wild potato resources [82].

Diploid potato has self-incompatibility [83] and male sterility [84]. Previous studies have shown that potato self-incompatibility is controlled by the “S-locus”. Hosaka and Hanneman [85] (1998) found that an S-locus suppressor gene (*Sli)* exists on chromosome 12 of diploid potato *Solanum chacoense*, which can overcome potato self-incompatibility. The seventh-generation diploid inbred line M6 was derived from *Solanum chacoense* and has high tuber dry matter content and strong disease resistance. In 2018, Leisner et al. [86] (2018) released the genome sequence of M6 and found that even though it was derived from a seventh-generation self-crossing population that was mostly homozygous, some regions still had high heterozygosity. The release of the M6 genome represented a significant step towards identifying genes related to important agronomic traits in inbred diploid potato.

In September 2020, Huang Sanwen’s team published the most complete potato heterozygous diploid genome to date. In this study, RH was used to complete the haploid assembly of heterozygous diploid potato for the first time, and the genome sequences of two sets of chromosomes were obtained. Analysis of this genome revealed rich sequence variation, allelic expression differences, methylation modification differences and the presence of harmful heterozygous mutations, providing critical information for breeding potato inbred lines [87].

In a recent genome study, the highly heterozygous tetraploid potato genome was decoded for the first time. In this study, Korbinian Schneeberger’s team sequenced the genome of a large number of single pollen cells and reconstructed a complete map of the haplotype genome of tetraploid potato, which represented a significant step forward in the breeding of tetraploid potato [88].

### 5.2. Potato Salt Tolerance Genes and Pathways

#### 5.2.1. Hormone Signals Involved in Response to Salt Stress in Potato

Plant hormones play important roles in growth, development and differentiation [89,90]. During abiotic stress, plant hormones mediate numerous signaling pathways to adapt to adverse conditions [91]. Thus far, nine plant hormones, including abscisic acid (ABA), auxin (IAA), gibberellin (GA), cytokinin (CK), ethylene (ET), brassinolactone (BRs), salicylic acid (SA), jasmonic acid (JA) and hornolactone (SLs), have been found to regulate plant growth in response to salt stress [92]. ABA is one of the most important stress hormones in plants. ABA has been found to increase rapidly under salt stress, and its response element (ABRE) binding protein and ABRE binding factor (AREB/ABF) transcription factors are phosphorylated by *SnRK2.2/2.3/2.6* [93]. In addition, ABA induces a decrease in stomatal area, the main site of plant transpiration, in response to salt stress. Salicylic acid levels also change in response to salt stress, leading to increased antioxidant synthesis, accumulation of osmotic compounds, and higher rates of photosynthesis [94]. BR induces the accumulation of NO, which can reduce the oxidative damage caused by salt stress through the clearance of reactive oxygen species (ROS) [95]. SL and ET mediate plant salt tolerance through their interaction with ABA signal transduction [92]. In general, plants reduce their growth rate in response to salt stress. Gibberellin, auxin and cytokinin inhibit growth and increase salt stress tolerance through signal transduction pathways [96,97,98]. The model plant *Arabidopsis thaliana* has been shown to activate the jasmonic acid signaling pathway during salt stress, thus inhibiting growth and increasing stress tolerance [99]. Additionally, the salicylic acid receptor NPR1 and its homologues NPR3 and NPR4 have been found to interact with salicylic acid to transmit signals during salt stress in potato. Li et al. [100] (2022) studied a member of the NPR family of proteins called NPR4 and found that the rooting rate of transgenic potato overexpressing the *StNPR4* gene increased significantly under high salt stress compared to the wild type, further confirming the role that *StNPR4* plays in salt stress adaptation. At present, no other hormone signaling pathways have been studied in potato salt stress (Figure 4).

#### 5.2.2. The Ca^2+^ Signaling Pathway Is Involved in Response to Salt Stress in Potato

Ca^2+^ is an important second messenger in plant cells, which has been shown to play key roles in plant environmental stress tolerance [101,102,103,104]. The majority of Ca^2+^ in plant cells is located around the cell wall and outer surface of the plasma membrane, where its concentration can reach more than 10,000 times that of the intracellular space [105]. Under salt stress, Na^+^ interacts with the receptor GIPCs outside the cell membrane, resulting in a change in the potential difference outside the cell membrane. This potential difference results in the opening of Ca^2+^ internal flow channels, causing a massive influx of extracellular Ca^2+^ and Ca^2+^ from the intracellular calcium bank into the cytoplasmic matrix, causing the intracellular Ca^2+^ concentration to increase rapidly and leading to the transmission of calcium signals [106,107,108,109]. Calcium receptors then decode calcium signals, including Ca^2+^ signal perception and Ca^2+^ signal transmission. Ca^2+^ receptor elements bind to Ca^2+^ produced by cells, transmit Ca^2+^ signals downstream through protein interactions or conformational changes, activate downstream elements and regulate downstream response genes to cope with salt stress. These genes are primarily divided into two main categories: Ca^2+^ response elements and Ca^2+^ sensing elements [110,111]. There are three main kinds of calcium ion receptors in plants: calcium-dependent protein kinase (CDPK), calmodulin protein (CaM) and calmodulin-like protein (CML), calcineurin B protein (CBL) and CBL-bound protein kinases (CIPK) [112]. The first two classes act within the CDPK signaling pathway and the CAM/CML signaling pathway. CDPK is a large Ca^2+^-sensitive family, which contains an N-terminal calmodulin-like regulatory domain that binds Ca^2+^ and causes conformational changes, releasing the self-inhibitory domain [113,114]. Hong et al. [115] (2020) identified 27 *StCDPK* genes. Among them, *StCDPK8/10/11/18/22* are related to salt stress response, and their expression levels are induced upon the perception of high salinity in order to improve tolerance to long-term stress. Zhu et al. [116] (2021) studied the mechanism by which *StCDPK32* participates in salt stress tolerance and found that *StCDPK32* protein is localized to the cell membrane, cytoplasm and nucleus. Salt treatment promotes the transcription of the *StCDPK32* gene, and overexpression of *StCDPK32* enhances the expression of salt-stress-related genes, increases the activity of stress-related enzymes and promotes plant photosynthesis in response to salt stress. Grossi et al. [117] (2022) found that plants overexpressing *StCDPK2* under high salt stress showed reduced peroxide accumulation and increased catalase activity, resulting in higher stress tolerance. CaM is a Ca^2+^-signal-sensing element, which binds to Ca^2+^ and undergoes a conformational change, and then transmits this signal to interacting proteins to drive signal transduction pathways under stress [118,119]. Raina et al. [120] (2021) found that *StCAM2* overexpression in tobacco significantly reduced reactive oxygen species accumulation and resulted in better photosystem II performance than wild plants under salt stress, leading to improved salt stress tolerance. The CBL/CIPK signaling pathway has been studied extensively in the model plant *Arabidopsis Thaliana*. Under salt stress, SOS3 (CBL4) first binds to Ca^2+^ and then forms a complex with SOS2 (CIPK24), thereby regulating SOS1’s interaction with intracellular Na^+^ and leading to reduced cell damage [121,122,123]. Thus far, no studies have been conducted to evaluate the CBL/CIPK signaling pathway during potato salt stress (Figure 4).

#### 5.2.3. Enzymes Involved in Response to Salt Stress in Potato

Many different genes encoding enzymes are involved in plant responses to salt stress. Sucrose non-fermentation-associated protein kinase (SnPK), for example, regulates plant growth and development and responds to abiotic stress through reversible protein phosphorylation [124,125]. Wang et al. [126] (2020) found that *StSnPK1* overexpression significantly increased SOD and POD activities as well as proline content under salt stress. Furthermore, the expression of genes related to proline biosynthesis, stress response and the active oxygen scavenging system were significantly upregulated, thus improving the salt tolerance of transgenic tobacco. Gao [127] (2015) examined the expression pattern of the *StSnPK2* gene under salt stress and found that the relative expression level of *StSnPK2.6* was significantly positively correlated with CAT and SOD activities. The *DWF4* gene encodes a C-22 hydroxylase, which is a rate-limiting enzyme for brassinosteroid (BR) synthesis [128]. Zhou et al. [129] (2018) cloned the *StDWF4* gene from potato and analyzed the phenotype of plants overexpressing this gene. Under salt stress, the MDA content of *StDWF4* overexpression mutant potato plants was lower than that of non-transgenic (NT) potato plants. Additionally, the contents of proline, soluble sugar and soluble protein, as well as the activities of antioxidant enzymes SOD, POD and APX, were higher than that of NT potato plants. Zhou [130] (2016) found that under salt stress, the relative gene expression of *StDWF4*-transformed potatoes increased significantly compared with NT potatoes, and the growth rate of potatoes was significantly higher than that of NT potato. Both with and without salt stress, the MDA content of transgenic potatoes was significantly lower than NT potato, and the contents of proline, soluble sugar and soluble protein were significantly higher than NT potato. Furthermore, the enzyme activities of SOD, POD, GR, CAT and APX were all much higher than those of NT potato. The ubiquitin-binding enzyme E2 is a key enzyme catalyzing the ubiquitin proteasome pathway that plays an important role in substrate ubiquitination [131]. Liu et al. [132] (2019) found that eight *StUBC* genes were responsive to salt stress based on qRT-PCR analysis, with *StUBC2*/*12*/*30*/*13* being the most highly expressed. Fu et al. [133] (2020) cloned the ubiquitin-binding enzyme E2 gene *StUBC12* from potato ‘Atlantic Ocean’ and found that overexpression of this gene enhanced potato salt stress tolerance (Figure 4).

#### 5.2.4. Transcription Factors Involved in Response to Salt Stress in Potato

Transcription factors are proteins that regulate gene transcription by binding to cis-acting elements of gene promoter regions to enhance or inhibit the transcription of target genes [134]. During abiotic stress, plants regulate the temporal and spatial expression of transcription factor genes at the transcriptional level in order to adapt to the stress [135,136]. To date, many transcription factors have been found to be involved in potato response to salt stress, including ERFs, NAC, DREB/CBF, TCP, WRKY, MYB and zinc finger proteins (Figure 4).

Studies have shown that ethylene response factor ERFs are involved in various biological processes associated with plant growth and development, biological stress and abiotic stress [137,138,139]. Wang et al. [140] (2021) conducted tissue-specific expression analysis of *StERF109* using qPCR. This analysis showed that *StERF109* was up-regulated under salt stress, with high expression seen in potato roots and variable expression changes in leaves and stems, implicating *StERF109* as a key gene in salt stress response. Charfeddine et al. [141] (2019) identified the *StERF94* gene and found that under salt treatment, the overexpression of *StERF94* improved the salt tolerance of potato by inducing osmoprotector synthesis. The NAC transcription factor family is involved in various stress responses, including salt stress in the model plants Arabidopsis [142], rice [143,144], soybean [145], wheat [146,147,148] and maize [149,150]. A total of 110 NAC transcription factor genes with different functions have been identified in potato [151]. Yue [152] (2021) found that under salt stress, overexpression of *StNAC1* increased the content of proline in tobacco, reduced the accumulation ROS and significantly enhanced the salt tolerance of plants. Yue et al. [153] (2021) showed that under salt stress, transgenic tobacco plants overexpressing *StNAC1* had higher seed germination rates, more green leaves, more proline accumulation and less ROS accumulation. Wang et al. [154] (2021) found that overexpression of the stress-induced *StNAC053* gene enhanced the tolerance of transgenic *Arabidopsis thaliana* to salt stress. Xu et al. [155] (2014) found that overexpression of *StNAC2* significantly enhanced the in vitro salt tolerance of potato (Figure 4).

The CBF/DREB protein is an important transcription factor in plants, which can specifically recognize CRT/DRE *cis*-acting elements in the downstream COR gene promoter, thus activating COR expression under abiotic stress to improve tolerance [156,157,158,159]. Song et al. [160] (2020) analyzed physiological and biochemical indexes of Arabidopsis plants overexpressing *StCBF1* and *StCBF4* under salt stress. The activities of antioxidant enzymes SOD, CAT, POD and APX were all increased, and fluorescence quantitative PCR analysis showed that *StCBF1* and *StCBF4* overexpression mutants had higher COR gene expression and Na^+^/H^+^ antiporter-related gene expression than the control wild-type *Arabidopsis Thaliana*. Bouaziz et al. [161] (2012) found that *StDREB2* overexpression in potato caused enhanced tolerance to salt stress. Bouaziz et al. [162] (2013) also isolated the *StDREB1* gene from potato and found that overexpression of *StDREB1* enhanced tolerance to salt stress. Additionally, *StDREB1* overexpression was found to activate a calcium-dependent protein kinase (CDPK) stress response gene. An in-depth study on the involvement of *StDREB1* in stress response found that the expression of a Cu/Zn SOD gene was enhanced in *StDREB1* transgenic plants under salt treatment [163]. TCP transcription factors are widely found in plants and regulate plant growth, development and environmental signals [164]. Li et al. [165] (2021) used *Escherichia coli* BL21 to express an sttcp13-Gst tag fusion protein and verified that the expression of the potato transcription factor *StTCP13* gene is induced by high salt stress and can improve the tolerance of *Escherichia coli* to high salt stress, indicating that *StTCP13* plays an important role in response to salt stress (Figure 4).

WRKY transcription factors regulate the expression of several gene families during stress response to attenuate cellular damage [166,167,168,169,170,171]. Yang et al. [172] (2020) used ‘Qingshu9’ potato as a test material, obtained the *StWRKY40* gene sequence by RT-PCR and analyzed the expression pattern of this gene under salt stress by qRT-PCR. *StWRKY40* was found to be responsive to salt stress in numerous tissues, peaking after four days of stress at nearly 20 times the level seen in unstressed plants. After eight days of salt stress, *StWRKY40* gene expression was still elevated in leaves. The involvement of MYB transcription factors in stress response has also been verified in rice [173,174], wheat [175,176], soybean [177,178,179] and other crops. Zhao [180] (2020) cloned the *StMYB2R-86* gene and showed no significant phenotypic differences between control wild-type potato and *StMYB2R-86*-overexpressing potato under normal growth conditions. Under salt stress, the height of potato plants overexpressing *StMYB2R-86* was significantly higher than that of the control, and the root system was more developed. Furthermore, physiological indexes indicated that *StMYB2R-86* expression in potatoes under salt stress resulted in more proline accumulation, lower malondialdehyde content, lower cell ion leakage rates and lower levels of superoxide O_2_^−^ in leaves (Figure 4).

Zinc finger proteins are transcription factors that play important roles in plant growth and stress responses [181]. Zhu et al. [182] (2021) cloned the *StZFP593* gene from ‘Qingshu 9’ and showed that it was induced in numerous tissues during salt stress. After five days of 100 mmol sodium salt treatment, *StZFP593* expression in roots and leaves was significantly up-regulated. After being treated with 150 mmol sodium for ten days, the expression level in roots and leaves peaked. Tian et al. [183] (2010) also found that ectopic expression of *StZFP1* enhanced the tolerance of transgenic tobacco to salt stress (Figure 4).

#### 5.2.5. Other Regulatory Factors That Participate in Response to Salt Stress in Potato

Protease suppressors (PIs) can inhibit the activity of proteases through steric hindrance. CYS is an inhibitor of cysteine proteases, which is involved in plant growth, development and stress responses [184]. Liu et al. [185] (2020) cloned and overexpressed the *StCYS1* gene in ‘Holland 15’ and found that, compared with wild potatoes, the transgenic potatoes accumulated more proline and chlorophyll content and had stronger H_2_O_2_ scavenging ability and cell membrane integrity during salt stress. CPI is a cysteine protease inhibitor that improves the resistance of many crops to abiotic stress [184,186]. Li [187] (2018) cloned the *StCPI* gene from tetraploid potato ‘Holland 15’ and found that *Escherichia-coli*-overexpressing *StCPI* could grow normally under high salt stress, and *StCPI* diploid potato had higher survival rates under high salt stress compared with wild-type.

The CCR4-NOT protein complex is a messenger between the nucleus and cytoplasm that is involved in mRNA synthesis and degradation, DNA damage repair and other processes. Rcd1 is the main subunit of the CCR4-NOT protein complex, and Chen et al. [188] (2019) cloned the *StRcd1* gene from potato ‘Qingshu 9’. Under salt stress, *StRcd1* gene expression changed in roots, stems and leaves and reaches its highest expression level at 12 h. Proline (Pro) is a major osmoregulatory substance, and proline transporters (ProT) belong to the Na^+^ -dependent amino acid transporter family [189]. Wang et al. [190] (2019) cloned and studied *StuProT1* and *StuProT2* proline transporter genes. After sodium salt stress, *StuProT1* and *StuProT2* were strongly induced at 2 h, with relative expression levels more than 19 times that of the control. In plants, sucrose is transported from source to reservoir through both symplast and extracellular pathways [191]. The apoplast pathway is mainly dependent on sucrose transporters [192,193]. Xie et al. [194] (2020) cloned the *StSUT2* sucrose transporter gene, which can be induced by salt stress in potato. The expression of this gene reached its maximum after eight hours of salt stress in roots and stems and after four hours of salt stress in leaves. The contents of proline, soluble sugar and soluble protein of tobacco-overexpressing *StSUT2* were also found to be higher than those of wild-type under salt stress, indicating enhanced salt stress tolerance (Figure 4).

Unlike most other crops, a major goal of potato salt tolerance research is to identify salt tolerance genes that also have a positive effect on the quality and yield of underground organs. The transformation from stolon to tuber is a dynamic process regulated by hormones, proteins and other factors [195]. Vitamin B6 is a cofactor in various metabolic reactions, and overexpression of PDX-II, a key vitamin B6 pathway gene in potato, has been shown to significantly increase vitamin B6 content in tubers [196]. In addition, these transgenic strains also showed better salt tolerance [197]. Sucrose plays a role in regulating the source–sink relationship and is a critical metabolic signal during tuber development. In plants, sucrose transporters transport sucrose from the ’source’ to the ’sink’ [198]. Reduction in sucrose transporter expression in potato has been shown to decrease fresh weight during early tuber development [199]. Additionally, the average fructose content in potato tubers overexpressing the *OsSUT5Z* gene was shown to be significantly increased, resulting in higher tuber yield and more tubers per plant [198]. ABF (ABRE binding factor) protein is a bZIP transcription factor that regulates abscisic acid signaling in response to abiotic stresses [200]. ABF proteins also play important roles in tuber formation [201]. Research has shown that the salt tolerance of Arabidopsis 35S: ABF4 transgenic potato was enhanced, and the tuber yield, storage capacity and processing quality of tubers were improved under both normal and stress conditions [202]. Potato *StPPI1* is a homologue of Arabidopsis proton-pump interaction factor 1, and its expression level increases during tuber development. Salt stress was also shown to increase the mRNA level of *StPPI1* [203]. Taken together, these results indicate that manipulating salt tolerance genes in potato also results in increased tuber quality and yield.

### 5.3. Application of Multiomics Approaches to Potato Salt Tolerance

With the continued development of high-throughput sequencing technology, multiomics has become a common technique to study abiotic stress in plants [204,205,206]. Multiomics provides a comprehensive view of molecular changes that take place during potato salt tolerance responses, including changes in gene expression, protein levels and metabolites. The application of these technologies has accelerated the identification of potato salt tolerance genes as well as the study of their biological functions. This approach not only helps to elucidate the molecular mechanism of potato response to salt stress, but also provides a theoretical basis and candidate genes for the genetic improvement of potato stress tolerance.

Earlier transcriptomic analyses of the response of potato to salt stress have involved the use of Illumina Hi Seq to conduct transcriptomic sequencing (NA-SeQ) on the tissue culture seedlings of salt-tolerant and sensitive cultivars after salt stress. After sequencing and gene expression analysis, differentially expressed genes of two potato cultivars under different salt stress conditions are then compared and assessed for gene ontology (GO) and KEGG enrichment. Key differentially expressed genes associated with salt tolerance can then be verified by real-time fluorescence quantitative PCR. Li [207] (2019) conducted transcriptome sequencing on the tissue culture seedlings of the salt-tolerant potato cultivar ‘Longshu 5’ and identified 20 key salt-tolerance-associated genes, which provided critical information for the cultivation of salt-tolerant potatoes. Proteomics is the large-scale study of the complete set of proteins expressed under a given set of conditions [208]. Metabolomics is one of the key omics techniques to study changes in plant metabolites during abiotic stress time courses [209]. This technique has been widely applied to salt stress in wheat [210], tomato [211] and maize [212]. Hamooh et al. [213] (2021) simulated a salt stress environment with LiCl, measured metabolite accumulation by GC-MS and spectrophotometry and compared the tolerance of red potato BARI-401 and yellow potato Spunta. Spunta accumulated more metabolites associated with stress tolerance during this time course, which may be correlated with its higher stress adaptability. Some research approaches aimed at understanding potato salt stress responses have combined transcriptomics with other high-throughput omics techniques. Evers et al. [214] (2012), for example, used transcriptomics and proteomics to comprehensively analyze the response of potato to cold stress and salt stress. At the transcriptomic level, the number of differentially expressed genes in potato under salt stress was less than that under cold stress, and genes related to primary metabolism, detoxification and signal transduction were strongly down-regulated. At the proteomic level, the number of differentially regulated proteins in potatoes increased by about three-fold under salt stress, and the level of most highly abundant proteins increased, except for genes associated with photosynthesis. Jing et al. [215] (2022) conducted transcriptome sequencing on potato DM plants and found 13 differentially expressed genes in the proline metabolism pathway that were correlated with increased proline content of DM plants.

## 6. Development and Utilization of Saline-Tolerant Potato Cultivars

It is important to develop salt-tolerant germplasm for tetraploid potato in order to increase potato yield. Khrais et al. [216] (1998) subjected the seedlings of 130 European and North American potato cultivars to NaCl stress, analyzed six growth parameters and conducted a cluster analysis. After this analysis, ‘BelRus’, ‘Bintje’, ‘Onaway’, ‘Sierra’ and ‘Tobique’ were identified as cultivars with strong salt tolerance. Since then, ‘Binjte’ has been widely used as a control cultivar for potato salt tolerance screening. Zhang et al. [217,218,219] (2010; 2013; 2014) used ‘Bintje’ as the control cultivar for salt tolerance screening of potato diploid clones in several analyses. Additionally, Liu et al. [220] (2010) conducted salt stress assays on 107 potato cultivars and determined that ‘Ningshu 5’ was a salt-tolerant cultivar. ‘Ningshu 5’, a late-maturing cultivar, is suitable for planting in the semi-arid mountainous area of south Ningxia in China. Zhang et al. [221] (2018) assessed morphological and physiological indexes during NaCl stress on potato cultivars from six different regions and found ‘Dongnong 09-33069’ to be a salt-tolerant cultivar. Zhou [222] (2018) studied the agronomic and quality traits of 18 potato cultivars grown on saline-alkali land and found that ‘M08’ had the smallest yield reduction and the highest overall yield. Based on these field experiments and assessment of seedlings treated with NaCl, ‘M08’ was determined to be a salt-tolerant cultivar. Additionally, Li et al. [223] (2018) determined ‘Longshu 5’, ‘LZ111’ and ‘391691.96’ to be salt-tolerant through salt stress treatment of 52 potato materials, measurement of physiological indexes and cluster analysis. ‘Longshu 5’ is a high-quality vegetable used for starch processing that is late-maturing. It is suitable for planting in the alpine region, the second Yin region and the semi-arid region of Gansu Province. ‘LZ111’ is a new potato strain suitable for flour processing with good yield, which is now a major processing potato in the Anding District, Gansu Province, China. Ahmed et al. [224] (2020) studied the morphological characteristics of tubercle and stolon growth of potato single-node stem slices under different concentrations of NaCl stress and determined that ‘Kennebec’ had strong salt tolerance by combining the results of a principal component analysis and hierarchical cluster analysis. Hu et al. [225] (2020) determined that the ‘Youjin885’ cultivar has strong salt tolerance by measuring the physiological characteristics of six potato cultivars under different NaCl concentrations. ‘Youjin 885’ is suitable for fresh food processing. It is suitable for planting in the northern one-season cropping area, the Central Plains two-season cropping area and high-altitude areas. Jiang et al. [226] (2021) determined that ‘Jinshu16’ and ‘Kexin19’ were moderately salt-tolerant cultivars based on the growth indicators of 11 potato cultivars under salt stress. ‘Jinshu 16’ is a medium- and late-maturing cultivar that is suitable for fresh potato export and starch processing, with high, stable yield and good adaptability. Yu et al. [227] (2022) comprehensively evaluated ‘Kexin23’, ‘Jizhangshu 3’ and ‘Jizhangshu 5’ cultivars and found that they have strong salt tolerance by measuring the changes of physiological indexes of ten potato cultivars subjected to salt stress and applying membership function analysis, cluster analysis and Grey correlation degree analysis. Among them, ‘Kexin23’ is an early-maturing fresh food cultivar. ‘Jizhangshu 3’ is a medium ripe cultivar bred in the Hebei Province of China, which is highly adaptable to the four main potato producing areas of China. It is suitable for sale in southern China, Hong Kong and Macao. ‘Jizhangshu 5’ is a medium cooked fresh potato cultivar suitable for growing in northern Hebei Province (Appendix A).

Diploid potatoes account for 73% of the potato germplasm resources, with significant room for improvement in salt stress tolerance [228]. Hawkes [229] (1990) divided cultivated potato into eight species, among which four species were diploid, including *Solanum stenotomum*, *Solanum phureja*, *Solanum goniocalyx* and *Solanum ajanhuiri*. *Solanum stenotomum* is one of the original species that is derived from other cultivated species, and *Solanum phureja* is used in breeding due to its high combining ability and ease of forming 2n gametes. In salt tolerance studies, hybrid asexual lines of *Solanum stenotomum* and *Solanum phureja* were screened for salt tolerance, and the screened salt-tolerant asexual lines were often used as parents along with salt-sensitive asexual material to construct salt-tolerant isolated populations for further screening and identification of new salt-tolerant cultivars. Zhang et al. [230] (2010) conducted salt tolerance screening on 45 in vitro seedlings of hybrid PHU-STN of diploid potato and found that the clone BD54-8 had high salt tolerance. After salt stress, the cell membrane, chloroplast and mitochondria of BD54-8 were still visible, and the ultrastructure of BD54-8 did not change significantly. Zhang et al. [231] (2013) conducted salt tolerance screening of 45 hybrid (PHU-STN) clones of diploid potato *Solanum phureja* and *Solanum stenotomum* and measured bud length, root length, bud fresh weight, root fresh weight, bud dry weight and root dry weight. Six clonal salt-tolerant materials (89-2-1, 188-1, 267-1, 566-1, 472-1 and 463-1) were identified from this work. Two salt-tolerant clones (A108 and A152) were selected by Ma [232] (2014) from the offspring of 472-1 (salt-tolerant) and 270-2 (salt-sensitive) parents, based on performance under salt stress. Zhao et al. [233,234] (2020; 2014) used 166 hybrid asexual lines of *Solanum phureja* and *Solanum stenotomum* (PHU-STN) as materials and obtained comprehensive evaluation D values by compounding the membership function and weight coefficients of six indicators: bud length, fresh weight of bud, dry weight of bud, root length, fresh weight of root and dry weight. They found that A108, A038, A024, A002, A152, A096, A003, A013 A200, A053, A066, A231, A139, A016, A120, A233, A254, A100, A255, A072, A050, A117, A238, A107, A069, A167 and A014 had better composite evaluation scores and were determined to have salt tolerance phenotypes. Wang [235] (2021) used the diploid salt-sensitive material HS66 and salt-tolerant material CE125 to construct salt-tolerant isolates and identified six highly salt-tolerant strains, including P3-403, P3-361, P3-563, P3-441, P3-482 and P3-394. Zaki and Radwan [236] (2022) examined the salt tolerance of diploid clones and found that PI 275136-6, PI 537025-8 and PI 566738-2 had the highest salt tolerance index in sodium salt screening. For agricultural production, the clones with high salt tolerance can be transplanted to the field and tested for 2n pollen production after flowering. After the identification of clones with 2n pollen, salt tolerance genes can be introduced into tetraploid cultivars by 4x-2x hybridization [237]. This scheme of using diploid potato salt-tolerant clones to breed tetraploid cultivars with high salt tolerance by decomposition breeding represents a novel way to develop salt-tolerant germplasm (Appendix A).

## 7. Future Perspectives

### 7.1. Improved Identification of Potato That Is Tolerant to Salt and Alkali Stress

Part of China’s saline-alkali land has the potential to be used for potato growing but requires the development of more tolerant cultivars. Previous studies have identified some cultivars that are tolerant to neutral salt, but there is still little information about basic salt tolerance. In recent years, researchers have focused more on the salt-alkali tolerance of diploid clones, and some studies suggest that the salt tolerance of diploid potato is related to alkali tolerance. However, more work is needed to determine whether a simi2016lar pattern of tolerance is present in tetraploid potato. The combined effects of alkali and salt conditions typically result in more dramatic yield reductions than stress alone [238,239]. Therefore, it is critical to conduct simultaneous saline-alkali screening of tetraploid potato populations in future research, in order to select and breed tetraploid potato cultivars that can be grown under these conditions.

### 7.2. Additional Exploration of Genes Associated with Salt Tolerance in Potato

In previous studies, the investigations of salt genes have mainly focused on identifying gene expression changes and physiological index changes, with little work performed on understanding the underlying mechanisms governing these changes. Future studies should therefore explore the mechanisms by which differentially expressed genes result in different levels of salt tolerance across potato germplasm, with the ultimate goal of creating more salt-tolerant potato cultivars.

## Figures and Tables

**Figure 1 ijms-24-10726-f001:**
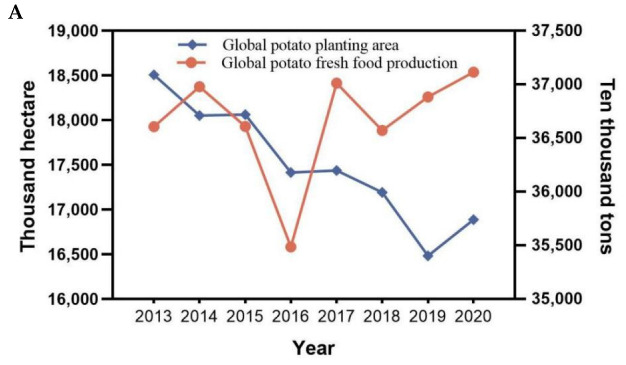
Potato planting area and fresh food production from 2013 to 2020. (**A**) Global potato planting area and fresh food production from 2013 to 2020. (**B**) Potato growing area and production in China from 2013 to 2020 (global potato planting area and fresh food production data source: https://www.fao.org/faostat/en/ (accessed on 15 April 2023); Chinese potato planting area and production data source: https://data.stats.gov.cn/ (accessed on 15 April 2023)).

**Figure 2 ijms-24-10726-f002:**
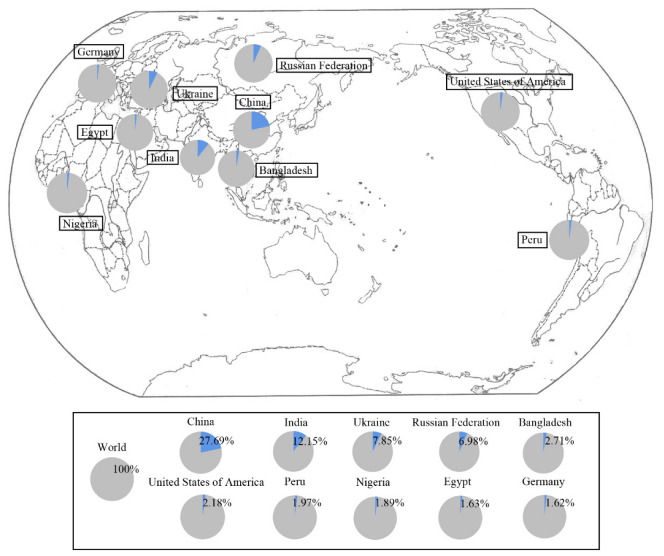
Proportion of potato growing area in the top ten countries in 2020 (data source: https://www.fao.org/faostat/en/ (accessed on 15 April 2023)).

**Figure 3 ijms-24-10726-f003:**
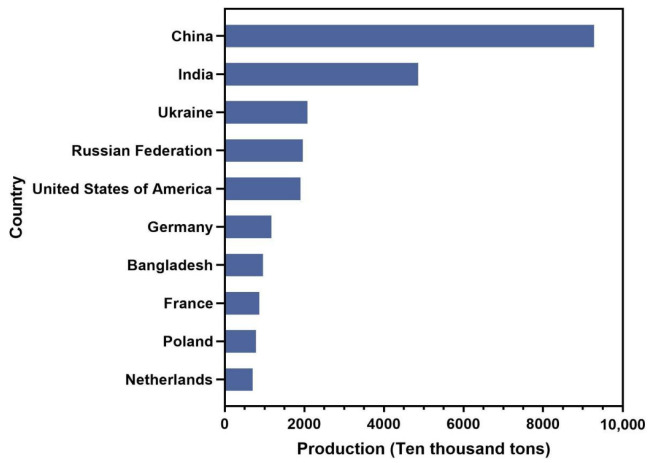
Fresh potato production in the top ten countries in 2020 (data source: https://www.fao.org/faostat/en/ (accessed on 15 April 2023)).

**Figure 4 ijms-24-10726-f004:**
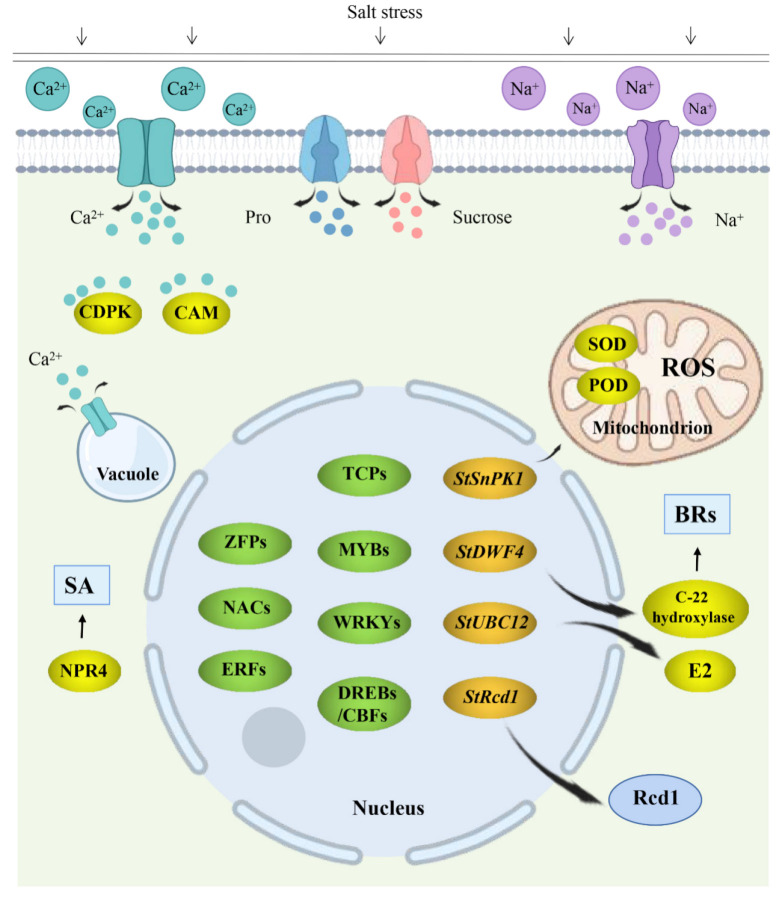
Proposed working model of salt tolerance genes and pathways in potato.

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
