# Peer review of "A Review of Potato Salt Tolerance"

_ijms, 2023, doi:10.3390/ijms241310726_

Round 1
Reviewer 1 Report
The manuscript under title “A review of potato salt and alkali tolerance” is interesting and presents many important information. However, in my opinion it should be improved.
The paper is very extensive and contains too many molecular details about signal transduction, transcription factors, or the action of the antioxidant system. In my opinion, this review paper should not include handbook information and detailed research results from other authors, but mainly the specific observations and comments on potato only. The biggest mistake in the work is the lack of a chapter on physiological, morphological and biochemical effects under salinity stress and soil alkalization. There is especially little information on alkalization stress. There are too many molecular reports, especially since this work shows that the mechanism of potato resistance to salt and alkalization stress is the same as that of other crops. The authors should focus primarily on the differences in potato resistance from other crops. For this reason, I suggest shortening the text and adding information on the negative effects of each stress.
Minor comments:
1. Latin names should be written in italics, e.g. Phytophthora infestans, Solanum commersonii, S. acoense etc. S. Commersonii should be replaced by S. commersonii.
2. suppressor gene, named Sli, should be: Sli
3. the authors write ‘varieties’everywhere. Varieties means botanical varieties while cultivated varieties are cultivars.
4. term ‘portion’ should be replaced by ‘parts’
Author Response
Dear Reviewers:
Thank you very much for your encouraging decision on our manuscript, “A review of potato salt and alkali tolerance”. Your feedback and suggestions have helped us to significantly improve the quality of the work. We have revised the MS carefully based on the suggestions from you.
We are pleased that our work was considered potentially acceptable for publication in IJMS, subject to major revisions. We thank you for the considerable time and effort you spent in reviewing the previous version of the manuscript. Based on the instructions provided in your letter, we have uploaded the revised manuscript files, with all changes highlighted using the track changes mode in MS Word.
Appended to this letter is our point-by-point response to the comments raised by you. The comments are reproduced in full, and our responses are given directly afterward.
We would also like to thank you for allowing us to resubmit a revised version of the manuscript. We deeply appreciate your consideration of our work, and we hope that the improved manuscript is deemed acceptable for publication in the International Journal of Molecular Sciences. If you have any questions, please do not hesitate to contact me.
Yours sincerely,
Reviewer 1:
The paper is very extensive and contains too many molecular details about signal transduction, transcription factors, or the action of the antioxidant system. In my opinion, this review paper should not include handbook information and detailed research results from other authors, but mainly the specific observations and comments on potato only. The biggest mistake in the work is the lack of a chapter on physiological, morphological and biochemical effects under salinity stress and soil alkalization. There is especially little information on alkalization stress. There are too many molecular reports, especially since this work shows that the mechanism of potato resistance to salt and alkalization stress is the same as that of other crops. The authors should focus primarily on the differences in potato resistance from other crops. For this reason, I suggest shortening the text and adding information on the negative effects of each stress.
Answer: Thank you for your advice. This is because there are very few studies on potato alkali resistance in China or worldwide. Based on the full text content and the opinions of the reviewers:
A: We deleted the information about alkali stress in section 2 and changed the title "A review of potato salt tolerance".
B: We have added a chapter on physiological, morphological and biochemical effects under salinity stress and have added information on the negative effects of salt stress.
C: We also added information regarding the difference between potato salt tolerance research and other crops.
Minor comments:
1.Latin names should be written in italics, e.g. Phytophthora infestans, Solanum commersonii, S. acoense etc. S. Commersonii should be replaced by S. commersonii.
Answer: Thank you for pointing out the mistake, we have modified it.
2.suppressor gene, named Sli, should be: Sli
Answer: Thank you for pointing out the mistake, we have modified it.
3.the authors write ‘varieties’everywhere. Varieties means botanical varieties while cultivated varieties are cultivars.
Answer: Thank you for pointing out the mistake, we have revised it based on your suggestions.
4.term ‘portion’ should be replaced by ‘parts’
Answer: Thank you for pointing out the mistake, we have modified it.
Reviewer 2 Report
The aim of the paper is reviewing progress towards the identification of saline-alkali tolerance genes in potato, covering the development and utilization of potato saline-alkali tolerant varieties. The topic of the paper is relevant due to a saline-alkali land is widely distributed across the word.
The scientific content of the manuscript is significant, but presented text does not fully cover the announced topic. This review presents only information about genes involved in response to salt stress in potato, none genes to alkali stress. Main content of paper is potato tolerance to salt stress and only two references concern to alkali stress of potato varieties. Current text of manuscript does not match the title.
References are not always appropriate:
Number 1 - Mahgoub, H.A.M.; Eisa, G.S.A.; Youssef, M.A.H. Molecular, biochemical and anatomical analysis of some potato (Solanum tuberosum L.) cultivars growing in Egypt. J. Genet. Eng. Biotechnol. 2015, 13, 39-49. https://doi.org/10.1016/j.jgeb.201 4.11.004
is inappropriate in the context of the global scale of potato crop.
Number 26 and 27 –
26. Kaur, B.; Gupta, S.R.; Singh, G. Soil carbon, microbial activity and nitrogen availability in agroforestry systems on moderately alkaline soils in northern India. Appl. Soil Ecol. 2000, 15, 283-294. https://doi.org/10.1016/S0929-1393(00)00079-
27. Zhang, X.; Wei, L.; Wang, Z.; Wang, T. Physiological and molecular features of Puccinellia tenuiflora tolerating salt and alkaline-salt stress. J. Integr. Plant Biol. 2013, 55, 262-276. https://doi.org/10.1111/jipb.12013.
are both inappropriate in the context of the annual crop yield loss and economic loss caused by salt and alkali stress.
Number 155 does describe A. thaliana, no wheat
The authors have made many mistakes in way of writing potato species S. chacoense, S. commersonii (see pages 5-6).
There is a strange error when authors cite the book of J.G. Hawkes (see page 12 the second paragraph). Indeed, Professor J.G. Hawkes named four cultivated potato species as Solanum phureja, Solanum stenotomum, Solanum goniocalyx, Solanum ajanhuiri, no other way.
Author Response
Dear Reviewers:
Thank you very much for your encouraging decision on our manuscript, “A review of potato salt and alkali tolerance”. Your feedback and suggestions have helped us to significantly improve the quality of the work. We have revised the MS carefully based on the suggestions from you.
We are pleased that our work was considered potentially acceptable for publication in IJMS, subject to major revisions. We thank you for the considerable time and effort you spent in reviewing the previous version of the manuscript. Based on the instructions provided in your letter, we have uploaded the revised manuscript files, with all changes highlighted using the track changes mode in MS Word.
Appended to this letter is our point-by-point response to the comments raised by you. The comments are reproduced in full, and our responses are given directly afterward.
We would also like to thank you for allowing us to resubmit a revised version of the manuscript. We deeply appreciate your consideration of our work, and we hope that the improved manuscript is deemed acceptable for publication in the International Journal of Molecular Sciences. If you have any questions, please do not hesitate to contact me.
Yours sincerely,
Reviewer2
The aim of the paper is reviewing progress towards the identification of saline-alkali tolerance genes in potato, covering the development and utilization of potato saline-alkali tolerant varieties. The topic of the paper is relevant due to a saline-alkali land is widely distributed across the word.
The scientific content of the manuscript is significant, but presented text does not fully cover the announced topic. This review presents only information about genes involved in response to salt stress in potato, none genes to alkali stress. Main content of paper is potato tolerance to salt stress and only two references concern to alkali stress of potato varieties. Current text of manuscript does not match the title.
Answer: Thank you for your advice. This is because there have been few studies conducted on potato alkali tolerance in China or worldwide. Based on the full text content and the opinions of the reviewers, we deleted the information about alkali stress and changed the title to "A review of potato salt tolerance".
References are not always appropriate:
Number 1 - Mahgoub, H.A.M.; Eisa, G.S.A.; Youssef, M.A.H. Molecular, biochemical and anatomical analysis of some potato (Solanum tuberosum L.) cultivars growing in Egypt. J. Genet. Eng. Biotechnol. 2015, 13, 39-49. https://doi.org/10.1016/j.jgeb.201 4.11.004
Answer: Thank you for pointing out the mistake. We have modified it and added the following reference:
Gebrechristos, H.Y.; Chen, W. Utilization of potato peel as eco-friendly products: A review. Food Sci. Nutr. 2018, 6, 1352-1356. https://doi.org/10.1002/fsn3.691.
is inappropriate in the context of the global scale of potato crop.
Number 26 and 27 –
- Kaur, B.; Gupta, S.R.; Singh, G. Soil carbon, microbial activity and nitrogen availability in agroforestry systems on moderately alkaline soils in northern India. Appl. Soil Ecol. 2000, 15, 283-294. https://doi.org/10.1016/S0929-1393(00)00079-
- Zhang, X.; Wei, L.; Wang, Z.; Wang, T. Physiological and molecular features of Puccinellia tenuiflora tolerating salt and alkaline-salt stress. J. Integr. Plant Biol. 2013, 55, 262-276. https://doi.org/10.1111/jipb.12013.
are both inappropriate in the context of the annual crop yield loss and economic loss caused by salt and alkali stress.
Answer: Thank you for pointing out this mistake. I have modified the text and added the following references:
- Barkla, B.J.; Castellanos-Cervantes, T.; de León, J.L.D.; Matros, A.; Mock, H.P.; Perez-Alfocea, F.; Salekdeh, G.H.; Witzel, K.; Zörb, C. Elucidation of salt stress defense and tolerance mechanisms of crop plants using proteomics--current achievements and perspectives. Proteomics 2013, 13, 1885-1900. https://doi.org/10.1002/pmic.201200399.
- Qadir, M.; Quillérou, E.; Nangia, V.; Murtaza, G.; Singh, M.; Thomas, R.J.; Drechsel, P.; Noble, A.D. Economics of salt‐induced land degradation and restoration. Resour. Forum 2014, 38, 282-295. https://doi.org/10.1111/1477-8947.12054.
Number 155 does describe A. thaliana, no wheat
Answer: Thank you for pointing out the mistake. I have modified it and added the reference:
Rahaie, M.; Xue, G.; Naghavi, M.R.; Alizadeh, H.; Schenk, P.M. A MYB gene from wheat (Triticum aestivum L.) is up-regulated during salt and drought stresses and differentially regulated between salt-tolerant and sensitive genotypes. Plant Cell Rep. 2010, 29, 835-844. https://doi.org/10.1007/s00299-010-0868-y.
The authors have made many mistakes in way of writing potato species S. chacoense, S. commersonii (see pages 5-6).
Answer: Thank you for pointing out the mistake. I have changed the writing and changed to italics.
There is a strange error when authors cite the book of J.G. Hawkes (see page 12 the second paragraph). Indeed, Professor J.G. Hawkes named four cultivated potato species as Solanum phureja, Solanum stenotomum, Solanum goniocalyx, Solanum ajanhuiri, no other way.
Answer: Thank you for pointing out the mistake. I have modified the text accordingly.
Reviewer 3 Report
Sufficient information about the previous studies have been collected and presented for readers to follow the present review rationale and procedures. The authors make a contribution to the research literature in this area of investigation. Moreover, the authors appropriately cited past literature. The manuscript is very well-written and flow logically. The review is thorough so the reader is given an adequate background about the topic. There was nothing major that I felt needed comment.
Author Response
Dear Reviewers:
Thank you very much for your encouraging decision on our manuscript, “A review of potato salt and alkali tolerance”. Your feedback and suggestions have helped us to significantly improve the quality of the work. We have revised the MS carefully based on the suggestions from you.
We are pleased that our work was considered potentially acceptable for publication in IJMS, subject to major revisions. We thank you for the considerable time and effort you spent in reviewing the previous version of the manuscript. Based on the instructions provided in your letter, we have uploaded the revised manuscript files, with all changes highlighted using the track changes mode in MS Word.
Appended to this letter is our point-by-point response to the comments raised by you. The comments are reproduced in full, and our responses are given directly afterward.
We would also like to thank you for allowing us to resubmit a revised version of the manuscript. We deeply appreciate your consideration of our work, and we hope that the improved manuscript is deemed acceptable for publication in the International Journal of Molecular Sciences. If you have any questions, please do not hesitate to contact me.
Yours sincerely,
Reviewer3
Sufficient information about the previous studies have been collected and presented for readers to follow the present review rationale and procedures. The authors make a contribution to the research literature in this area of investigation. Moreover, the authors appropriately cited past literature. The manuscript is very well-written and flow logically. The review is thorough so the reader is given an adequate background about the topic. There was nothing major that I felt needed comment.
Thank you very much for your encouraging decision on our manuscript.
Round 2
Reviewer 1 Report
The manuscript has been revised and is ready for publication in its present form